# Effects of Deuterium Depletion on Age-Declining Thymopoiesis In Vivo

**DOI:** 10.3390/biomedicines12050956

**Published:** 2024-04-25

**Authors:** Nataliya V. Yaglova, Sergey S. Obernikhin, Ekaterina P. Timokhina, Dibakhan A. Tsomartova, Valentin V. Yaglov, Svetlana V. Nazimova, Elina S. Tsomartova, Marina Y. Ivanova, Elizaveta V. Chereshneva, Tatiana A. Lomanovskaya

**Affiliations:** 1Laboratory of Endocrine System Development, A.P. Avtsyn Research Institute of Human Morphology of Federal State Budgetary Scientific Institution “Petrovsky National Research Centre of Surgery”, 119991 Moscow, Russia; lesd@morfolhum.ru (S.S.O.); rodich@mail.ru (E.P.T.); dtsomartova@mail.ru (D.A.T.); vyaglov@mail.ru (V.V.Y.); pimka60@list.ru (S.V.N.); tselso@yandex.ru (E.S.T.); 2Department of Human Anatomy and Histology, Federal State Funded Educational Institution of Higher Education I.M. Sechenov First Moscow State Medical University, 119435 Moscow, Russia; ivanova_m_y@mail.ru (M.Y.I.); yelizaveta.new@mail.ru (E.V.C.); tatiana_80_80@inbox.ru (T.A.L.)

**Keywords:** deuterium depletion, thymus, T cells, proliferation, differentiation, migration

## Abstract

The thymus provides maturation and migration of T cells to peripheral organs of immunity, where they recognize diverse antigens and maintain immunological memory and self-tolerance. The thymus is known to be involved with age and in response to stress factors. Therefore, the search for approaches to the restoration of thymopoiesis is of great interest. The present investigation was aimed at evaluating how prolonged deuterium depletion affects morphogenetic processes and the physiological transition of the thymus to age-related involution. The study was performed on 60 male Wistar rats subjected to consumption of deuterium-depleted water with a 10 ppm deuterium content for 28 days. The control rats consumed distilled water with a normal deuterium content of 150 ppm. The examination found no significant differences in body weight gain or the amount of water consumed. The exposed rats exhibited similar to control dynamics of the thymus weight but significant changes in thymic cell maturation according to cytofluorimetric analysis of thymic subpopulations. Changes in T cell production were not monotonic and differentially engaged morphogenetic processes of cell proliferation, differentiation, and migration. The reactive response to deuterium depletion was a sharp increase in the number of progenitor CD4^−^CD8^−^ cells and their differentiation into T cells. The compensatory reaction was inhibition of thymopoiesis with more pronounced suppression of differentiation of T-cytotoxic lymphocytes, followed by intensification of emigration of mature T cells to the bloodstream. This period lasts from 3 to 14 days, then differentiation of thymic lymphocytes is restored, later cell proliferation is activated, and finally the thymopoiesis rate exceeds the control values. The increase in the number of thymic progenitor cells after 3–4 weeks suggests consideration of deuterium elimination as a novel approach to prevent thymus involution.

## 1. Introduction

The thymus and red bone marrow are the central organs of lymphocytopoiesis. Thymus function is unique since other organs of the immune system are not capable of producing T lymphocytes.

T cells originate from bone marrow early progenitors that migrate to the thymus. The thymus provides the proliferation, differentiation, selection, and emigration of mature T cells to the bloodstream and peripheral organs of immunity [1,2].

T cells recognize bacterial, viral, and tumor antigens and maintain immunological memory and self-tolerance. They carry out adaptive immune reactions and regulate their intensity throughout life. Therefore, the maintenance and stimulation of cell renewal in the thymus have attracted the interest of many researchers. In addition to its unique function, the thymus has physiological characteristics that make it a more vulnerable part of the immune system than the bone marrow and secondary lymphoid organs. A peculiarity of the thymus is its early age-related involution, which begins after puberty [3,4]. Besides age-dependent changes, the thymus is known to undergo involution in response to stress, infection, negative environmental factors, radiotherapy, and chemotherapy [5,6,7,8,9]. Thymus depletion has been reported to result in immunosenescence, inflammation, and impaired T-cell-mediated immunity [10]. The thymus has been shown to regenerate after injury, but its regenerative capacity also depends on age [11,12]. Emergent methods of thymus restoration like transplantation of T cell progenitors, transplantation of thymic tissue, thymic organoids, and artificial thymus transplantation are considered effective but expensive and laborious techniques [13,14]. Alternative strategies include hormonal and cytokine therapy, which also have limitations [15,16,17]. The development of methods for the prevention of thymus involution is an equally difficult problem to solve. Maintenance of thymopoiesis strictly depends on the activity of thymic epithelial cells. Therefore, the search for new approaches to regeneration of thymic cell populations and restoration of thymopoiesis in adults requires comprehensive study and identification of factors that can support self-renewal of both thymic subpopulations, lymphocytes, and stromal cells. Among such promising factors is the changing balance of stable isotopes in biogenic atoms.

Deuterium, a stable isotope of hydrogen, is abundant in living nature. The proportion of hydrogen isotope protium is 99.985% and deuterium is 0.015% [18]. However, deuterium content in mammals is high and second only to sodium. Its concentration in the blood is many times higher than that of the essential macroelements potassium, calcium, and magnesium [19]. Since biochemical reactions involving protium and deuterium have different rates, the substitution of protium by deuterium in chemical substances significantly increases the bond breaking energy and slows down the relative rate of chemical reactions [20]. These properties of deuterated and deuterium-depleted substances provide the basis for considering the hydrogen isotope balance as a putative regulator of metabolic and physiological processes.

Altered deuterium supply has been shown to provoke shifts first in deuterium/protium balance and then in cytosol and biopolymers [19]. Elimination of deuterium seems to induce diverse effects in normoblastic and tumor cells. Prolonged elimination of deuterium has been reported to substantially alter tumor cell fate [21,22,23]. Some investigations have shown that incubation of malignant cells in deuterium-depleted water for 48 h results in oxidative stress in the cells, which activates apoptosis and downregulates the expression of genes regulating cell survival [24,25]. Reduction of the heavier hydrogen isotope in normal conditions triggers an immediate response consisting of intensified secretory and morphogenetic processes [26,27]. Later effects of deuterium depletion on normoblastic cells are less studied. Our investigation aimed at the evaluation of thymic lymphocyte production in normal conditions in adolescent rats in order to assess how prolonged deuterium depletion induced by consumption of deuterium-depleted water affects the physiological transition of the thymus from a period of maximum development and functional activity to an age-related involution. We focused on the dynamic evaluation of thymoblast differentiation and migration of mature T cells to the bloodstream, the two main morphogenetic processes that provide thymic cell output, and reactive and compensatory changes in thymopoiesis to deuterium depletion.

## 2. Materials and Methods

### 2.1. Animals and Experimental Design

Male Wistar rats aged 8 weeks, weighing 210–230 g, (*n* = 60), were purchased from the Scientific Center of Biomedical Technologies of Federal Medical and Biological Agency of Russia (Pushchino, Moscow region, Russia). The rats were housed (4–5 rats in cage) at +22–23 °C with a 12/12 h light–dark cycle and given a pelleted standard chow and water ad libitum. The rats were enrolled in the experiment after two weeks of adaptation to the local vivarium.

The rats were randomized into two groups. The control group (*n* = 30) consumed distilled tap water with a normal deuterium content (150 ppm). The experimental group (*n* = 30) consumed deuterium-depleted water with [D] = 10 ppm ad libitum instead of tap water for 28 days. Deuterium-depleted water was manufactured by the B. P. Konstantinov St. Petersburg Nuclear Physics Institute of National Research Center “Kurchatov Institute” (St. Petersburg, Russia). The concentration of deuterium in water samples was verified using the T-LWIA-45-EP isotope analyzer (Los Gatos Research Inc., San Jose, CA, USA), which determines deuterium content with an accuracy of 1 ppm. The body weight of the rats and the volume of the consumed water were measured daily for terms 1 and 3 days, then every 3 days for terms 7, 14, 21, and 28 days. The amount of water consumed per 1 g of body weight was calculated. The rats were sacrificed on the 1st, 3rd, 7th, 14th, 21st, and 28th days by zoletil overdosage. The thymus was removed and weighed. The relative weight of the organ was calculated as a percent of body weight.

### 2.2. Thymic Lymphocytes Culture Preparation

The thymus was homogenized to obtain a culture of thymocytes in RPMI 1640 medium. Lymphocytes were separated from the stroma, including the thymic epithelial cells, by squeezing through meshes with 40 μm holes. The cell suspension was washed twice by centrifugation in the same medium for 5 min at 1000 rpm, bringing the concentration to 10 million cells in 1 mL.

### 2.3. Flow Cytometry

The subpopulation of thymic lymphocytes was assessed by flow cytometry using antibodies to CD3, CD4, and CD8 conjugated with fluorochromes (Invitrogen, Carlsbad, CA, USA). The sample preparation procedure was carried out according to standard protocols; an FC500 flow cytometer (Beckman Coulter, Brea, CA, USA) was used for the study. The percentage of CD3-positive lymphocytes was determined, including single-positive T-helpers (CD3^+^CD4^+^CD8^−^) and T-cytotoxic (CD3^+^CD4^−^CD8^+^) cells, as well as double-positive (CD3^−^CD4^+^CD8^+^) and negative (CD3^−^CD4^−^CD8^−^) cells representing lymphoblasts.

### 2.4. Ethical Approval

The investigation was performed in accordance with the handling standards and rules of laboratory animals as consistent with “International Guidelines for Biomedical Researches with Animals” (1985), in accordance with GOST 33215-2014 (Guidelines for Accommodation and Care of Animals. Environment, Housing and Management) and GOST 33216-2014 (Guidelines for Accommodation and Care of Animals. Species-Specific Provisions for Laboratory Rodents and Rabbits) and laboratory routine standards in the Russian Federation (Order of Ministry of Healthcare of the Russian Federation dated 19.06.2003 No. 267) Animal experiments were approved by the Ethics Committee of the Research Institute of Human Morphology (protocol No. 23, 25 March 2020).

### 2.5. Statistical Analysis

Statistical analyses were carried out using the software package “Statistica 7.0” (StatSoft, Tulsa, OK, USA). Normality of distribution was conferred by the Shapiro–Wilk test. The central tendency and dispersion of quantitative traits with an approximately normal distribution were presented as the mean and standard deviation (M ± S.E.M.). Quantitative comparisons of independent groups were performed using the Student’s *t*-test, taking into account the values of Levene’s test for the equality of variances. Quantitative comparisons of lymphocyte subpopulation percentages were performed using Chi-square. Differences were considered statistically significant at *p* < 0.05.

## 3. Results

### 3.1. Body Weight Changes during the Experiment

A progredient increase in body weight was observed in the control group since young rats were enrolled in the experiment. The rats that consumed deuterium-depleted water also demonstrated a gain in body weight. No significant differences in body weight between the control and the exposed animals were found at each time point of the experiment (Figure 1).

### 3.2. Parameters of Water Consumption

The water consumption curve was wavy for the control rats. Consumption of deuterium-depleted water demonstrated the same trend and the absence of statistically significant differences between the control and the exposed rats (Figure 2).

### 3.3. Thymus Weight

Absolute and relative thymus weight showed an increase up to the 7th day of the experiment in the control rats (Figure 3A,B). No significant changes were found in both parameters from the 7th to the 28th days of the investigation. The rats that consumed deuterium-depleted water demonstrated the same changes in the absolute and relative thymus weight, but the maximal values were observed on the 14th day of the experiment (Figure 3A,B). No statistically significant differences in thymus weight values between the control and the exposed rats were found.

### 3.4. Evaluation of Thymic Lymphocyte Subpopulations

#### 3.4.1. CD4^−^CD8^−^ Progenitors

Double-negative CD4^−^CD8^−^ cells represented the smallest subpopulation of thymic lymphocytes. Its percentage demonstrated some fluctuations during the first two weeks of the investigation and a significant decrease by the end of the experiment in the control (Figure 4).

The rats that consumed deuterium-depleted water exhibited an altered pattern of early progenitors’ content in the thymus. A significant increase in the percentage of CD4^−^CD8^−^ cells was observed 1 day after deuterium depletion began. Further changes were associated with a gradual reduction in double-negative cells for two weeks and a second, smaller increase on the 21st and 28th days. That is why the values of the deuterium-depleted rats were significantly different from the control at each time point (Figure 4).

#### 3.4.2. CD4^+^CD8^+^ Lymphocytes

Double-positive cells were abundant in the thymus. The percentage of CD4^+^CD8^+^ differentiating lymphocytes in the control rats also had fluctuations, with minimal values on days when the number of double-negative cells was high (Figure 5).

The exposed rats showed a significant reduction in double-positive cells after 1 day of consumption of deuterium-depleted water. Further, they exhibited a higher rate of CD4^+^CD8^+^ lymphocytes than the control animals (Figure 5). The amplitude of fluctuations in double-positive cell content was also smaller than in the control rats.

#### 3.4.3. CD3^+^ Lymphocytes

The percentage of CD3^+^ lymphocytes representing mature T cells in the control group was characterized by moderate fluctuations throughout the study period (Figure 6).

The rats that consumed deuterium-depleted water exhibited a significantly higher amplitude of fluctuations due to multiple expansions of CD3-positive cells one day after the beginning of the exposure. Subsequently, the rats showed a decrease in CD3^+^ cell content in the thymus, but on the 14th day of deuterium-depleted water consumption, the percentage of mature T cells again exceeded the control values. Then, the T cell content decreased again and, by the end of the experiment, corresponded to the values of the control group (Figure 6).

#### 3.4.4. T-Helper CD3^+^CD4^+^ and T-Cytotoxic CD3^+^CD8^+^ Lymphocytes

The content of CD3^+^CD4^+^ cells in the control thymus varied during the investigation, but was in correspondence with fluctuations in CD3^+^ cell percentage (Figure 7A). CD3^+^CD8^+^ lymphocytes demonstrated a higher amplitude of fluctuations than CD3^+^CD4^+^ cells (Figure 7B). The T-helper/T-cytotoxic lymphocyte ratio also ranged, but always was more than one (Figure 7C).

Deuterium depletion induced significant changes both in the maturation of T-helper and T cytotoxic lymphocytes (Figure 7A,B). The contribution of T-helper and T-cytotoxic lymphocytes to the surge in T-cell content one day after the onset of exposure was approximately equal. A further decrease in T-cell numbers was associated mainly with a lower content of T-cytotoxic lymphocytes in the thymus. These changes produced a disbalance in the T-helper/T-cytotoxic lymphocyte ratio (Figure 7C). On the 28th day of the exposure, the percentages of the CD3^+^CD4^+^ and CD3^+^CD8^+^ subpopulations were within the control values.

### 3.5. T Cell Efflux to the Blood

Quantification of CD3-positive lymphocytes in blood samples revealed another type of curve of the T cell content in blood and a lower amplitude of fluctuations in the exposed rats. After one day of exposure, the percentage of T cells was significantly reduced compared to the control. No significant fluctuations in T cell percentage were observed from the 3rd to the 21st day. On the 28th day, a significant increase in T cell blood content was registered (Figure 8).

## 4. Discussion

The present investigation showed that consumption of deuterium-depleted water did not induce pronounced physiological changes in rats but significantly altered morphogenetic processes in the thymus. The evaluation of body weight and the amount of water consumed showed that deuterium depletion did not affect metabolic rate. The exposed rat and the control rat similarly increased their weight. Examination of the control thymus revealed its enlargement with the maximum absolute and relative weight on the 7th day of the investigation and no further growth, indicating a transition of the thymus from growth to involution. The rats that consumed deuterium-depleted water demonstrated the same trend but with a little bit lower rate of thymus growth and maximal weight values on the 14th day. It suggests that the revealed differences were associated with some changes provoked by deuterium depletion.

The present results indicate that consumption of deuterium-depleted water evoked some alterations in thymopoiesis, which requires thorough analysis. Production of T cells in the thymus includes the emigration of early pre-T cell progenitors from the red bone marrow and their proliferation and renewal in the subcapsular region of the thymus cortex [2]. Then early CD4^−^CD8^−^ progenitors differentiate into double-positive CD4^+^CD8^+^ cells. Double-positive cell fate is associated with loss of expression of one of the clusters of differentiation and parallel activation of expression of the CD3 molecule. Further single-positive CD3^+^CD4^+^ or CD3^+^CD8^+^ T cells migrate from the cortex to the medulla, where they immigrate from the thymus to the blood [28,29]. Consequently, the production of T cells in the thymus represents tight cooperation of the morphogenetic processes—cell proliferation, differentiation, and migration. To more accurately assess changes in these three processes, it is useful to evaluate their contribution to T cell formation at each study time point. The control rats demonstrated a wavelike pattern of T cell output. It corresponds with reported data on infradian rhythms of morphological and functional changes in the thymus [30]. Consumption of deuterium-depleted water significantly changed T cell production, but the changes were not linear and showed no direct association with a level of deuterium depletion. Consumption of deuterium-depleted water has been shown to decrease deuterium blood content first and later reduce deuterium concentrations in solid tissues [31,32].

According to these data, consumption of deuterium-depleted water during one day alters the protium/deuterium balance in the blood and the lymph. This shift provoked the most pronounced changes in thymopoiesis. It confirms previously obtained data on the rapid response of the thymus to deuterium depletion [27]. A proportional increase in nondifferentiated CD4^−^CD8^−^ and mature CD3^+^ cells indicates that the balance between CD3^+^ and early progenitors is not disturbed. Such a strong increase in T cells suggests the accelerated differentiation of double-positive to single-positive cells. This is also evidenced by the decrease in the percentage of double-positive cells among thymic lymphocytes. The second reason for the increase in the percentage of differentiated T cells was their nonincreasing emigration from the thymus into the systemic bloodstream. However, activation of proliferation and differentiation of progenitor cells was obviously the main cause of T cell accumulation. Noteworthy is that deuterium depletion stimulated differentially the differentiation of T-helper and T-cytotoxic lymphocytes, increasing the maturation of the former to a greater extent.

On day 3, we observed reversed changes in the number of lymphocyte subpopulations in the thymus, such as a decrease in the percentage of undifferentiated and highly differentiated cells and a decrease in the number of differentiating double-positive cells. The T-cell blood count was also reduced. These findings indicate inhibited T cell production, which is most likely the cause of functional depletion and exhaustion of the early progenitor pool of cells after such a dramatic increase in morphogenetic activity. Like the first day of the study, inhibition of differentiation of T cell subpopulations was also different. The suppression of differentiation affected mainly T-cytotoxic lymphocytes.

On the 7th day of deuterium depletion, the content of mature T cells in the thymus remained decreased, while in the blood, it exceeded the control values. The decrease affected both T-helper and T-cytotoxic lymphocytes. The percentage of low-differentiated CD4^−^CD8^−^ cells was significantly reduced. At the same time, the content of double-positive cells was higher than in the control. That is, we observed an enhanced migration of T cells from the thymus into the bloodstream.

On the 14th day of the study, the number of T cells in the thymus increased and exceeded the control values. At the same time, the content of undifferentiated cells was three times lower than in the control, and the percentage of double-positive cells was a little bit higher, indicating accelerated differentiation of cells. The content of T-helper cells completely normalized, while that of T-cytotoxic cells remained reduced. The number of T cells in the blood corresponded to the control values, which confirms the activation of the cell differentiation process.

On the 21st day, the content of differentiated T cells in the thymus and blood decreased. Differentiation processes were characterized by the increased formation of T-cytotoxic lymphocytes. Evaluation of thymic lymphocytes revealed an increased number of low-differentiated cells, which, together with an adequate proportion of double-positive cells, indicates the activation of progenitor cell proliferation.

After 28 days of consumption of deuterium-depleted water, the content of T cells in the thymus corresponded to the control values, and in the blood, it exceeded them by a quarter. It indicates increased emigration of mature T lymphocytes from the thymus. The number of low-differentiated cells in the thymus exceeded the values of the control group more than three times. The rates of differentiation of low-differentiated cells into double-positive cells corresponded to normal values. Taken together, these findings denote the activation of the proliferation of T-cell precursors. We also observed a restoration of balance between T-helper and T-cytotoxic lymphocytes in the thymus.

Thus, the obtained data show that a decrease in deuterium intake into the organism causes significant changes in rates of morphogenetic processes in the thymus. The reaction observed after a day of consumption of deuterium-depleted water shows that deuterium content in the systemic bloodstream is a factor regulating cell proliferation and differentiation. Deuterium depletion stimulated the formation of an early progenitor pool of cells in the thymus. It is important to note that there was no long-term persistence of CD4^−^CD8^−^ cells in the thymus, mainly due to their enhanced differentiation. It should be considered a favorable factor since the long-term accumulation of low-differentiated cells in the thymus is known as a risk factor for lymphoma development [33]. However, it should be noted that deuterium reduction did not increase the migration of mature T cells into the systemic bloodstream. That is, of the three fundamental processes of thymopoiesis, deuterium concentration clearly contributed to only two: the generation of thymoblasts and their differentiation. Further observed changes indicated a decrease in T-cell production due to the depletion of the content of early thymic precursors. However, the differentiation processes of the existing immature cells were not slowed down. The restoration of the number of undifferentiated cells and their active differentiation in the thymus after a month of exposure, when the change in deuterium content occurs not only in blood but also in tissues, once again show that T cell maturation is a deuterium-dependent process. It is known that deuterium reduction has an antioxidant effect on cells and increases their sensitivity to hypoxia [34]. This is an important factor in increasing cell survival. Many of these effects are explained by changes in mitochondrial functioning, that is, the activation of cellular respiration [25]. However, this does not explain exactly why differentiation processes are enhanced. A probable explanation for the activation of the synthesis of key molecules CD3, CD4, and CD8 in thymocytes may be the activation of transcription of the corresponding genes due to the change in the number of open states in the DNA molecule. The recent studies have highlighted the formation of the DNA open states as being responsible for the «recognition» of the transcription start point and, accordingly, as an earlier event, if compared with the instant of binding an enzyme to its specific site [35,36,37]. A number of studies have shown that a decrease in the deuterium content and replacement of deuterium by protium in the DNA molecule reduces the energy of breaking complementary hydrogen bonds between pairs of bases, that is, contributes to an increase in the number of open states in genes [38,39]. Such changes in the structure of cellular DNA can be considered a factor that increases the possibility of gene transcription and the activation of respiration in cells, as well as a factor that contributes to the enhancement of energy supply for the processes of mRNA translation and protein synthesis. In our previous investigation, we found a strong stimulatory effect of 24 h consumption of deuterium-depleted water on the secretion of thyroid hormones, which are known to promote differentiation of thymic lymphocytes [26]. These findings allow us to consider them an additional mechanism of activated cell differentiation. The 2-week consumption of deuterium-depleted water with a D concentration of 10 ppm has been found to suppress pituitary thyroid stimulating hormone and consequently thyroid hormone secretion, but a longer reduction in deuterium content did not affect thyroid function [40]. The present investigation demonstrates enhanced differentiation of thymic progenitor cells on the 14th day of deuterium depletion, suggesting independent pituitary–thyroid axis mechanisms of active cell maturation. It is especially interesting to note that the reduction in deuterium supply did not equally enhance the differentiation of double-positive cells. The formation of CD3^+^CD4^+^ T-helpers was more active than CD3^+^CD8^+^ T cytotoxic cells after a day of exposure. Cytotoxic lymphocytes are an important link in the anti-infectious and antitumor responses. Peripheral T cytotoxic cells are known to exhaust [41,42]. It significantly affects the elimination of virally transfected and malignant cells [43,44]. Since no significant release of T-lymphocytes into the systemic bloodstream was observed after a day of consumption of deuterium-depleted water, it means that an additional pool of naive T-cytotoxic cells was formed in the thymus, which could migrate to the sites of the pathological process. Exhaustion of T cytotoxic cells in tumors is crucial for antitumor immunity. Since the generation of T cells in the thymus is attenuated with age, deuterium depletion evokes some effects, which suggests a novel strategy for rejuvenation of the thymus in adults. During the period of thymopoiesis depletion, the ratio of T-cytotoxic and T-helper cells gradually shifted to control values, and, consequently, a long-term reduction in deuterium content in the body does not cause significant changes in the balance of T-helper and T-cytotoxic lymphocytes either in the thymus or in the blood.

## 5. Conclusions

Morphogenetic processes, which provide thymopoiesis, are sensitive to deuterium content in the blood and solid tissues. Deuterium depletion attenuates the transition of the thymus to age-related involution and, in general, maintains a higher rate of thymopoiesis. Changes in T cell production are not monotonic and differentially engage morphogenetic processes of cell proliferation, differentiation, and migration. The reactive response to deuterium depletion is a sharp increase in the number of progenitor cells and their proliferation and differentiation into T cells. The compensatory reaction is inhibition of thymopoiesis with more pronounced suppression of differentiation of T-cytotoxic lymphocytes, followed by intensification of emigration of mature T cells to the bloodstream. This period lasts from 3 to 14 days, then differentiation of thymic lymphocytes is restored, and later, progenitor cell proliferation is activated. T-cytotoxic lymphocytes are more sensitive to deuterium content in the body. Normalization of their differentiation occurs at 21 days.

The increase in the number of thymoblasts after 3–4 weeks and higher parameters of thymopoiesis after 28 days suggest consideration of deuterium elimination as a novel approach to prevent thymus involution.

### Limitations of the Study

The results of the investigation can depend on age.

## Figures and Tables

**Figure 1 biomedicines-12-00956-f001:**
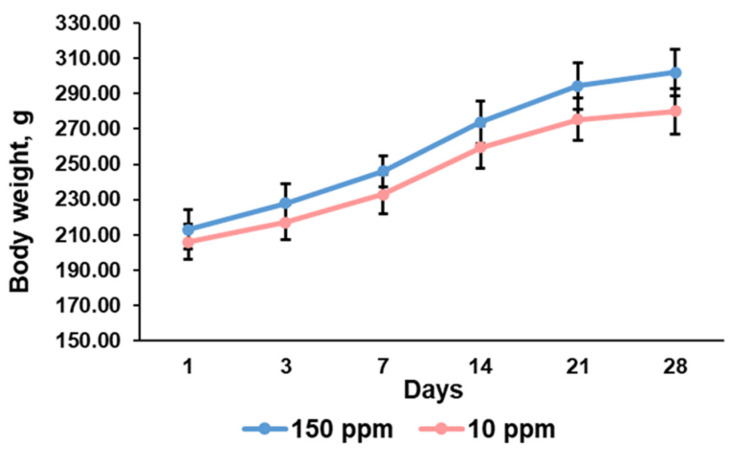
Dynamics of body weight of the control rats (150 ppm) and the rats that consumed deuterium-depleted water (10 ppm). Data are shown as mean ± S.E.M.

**Figure 2 biomedicines-12-00956-f002:**
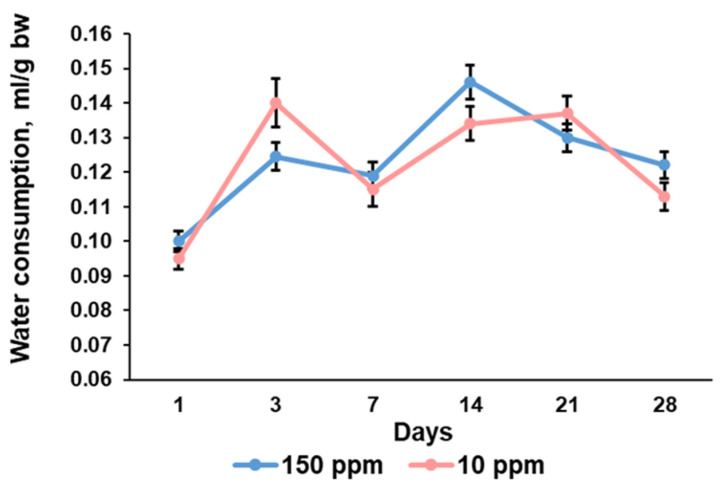
Dynamics of water consumption per 1 g of body weight of the control rats (150 ppm) and the rats that consumed deuterium-depleted water (10 ppm). Data are shown as mean ± S.E.M.

**Figure 3 biomedicines-12-00956-f003:**
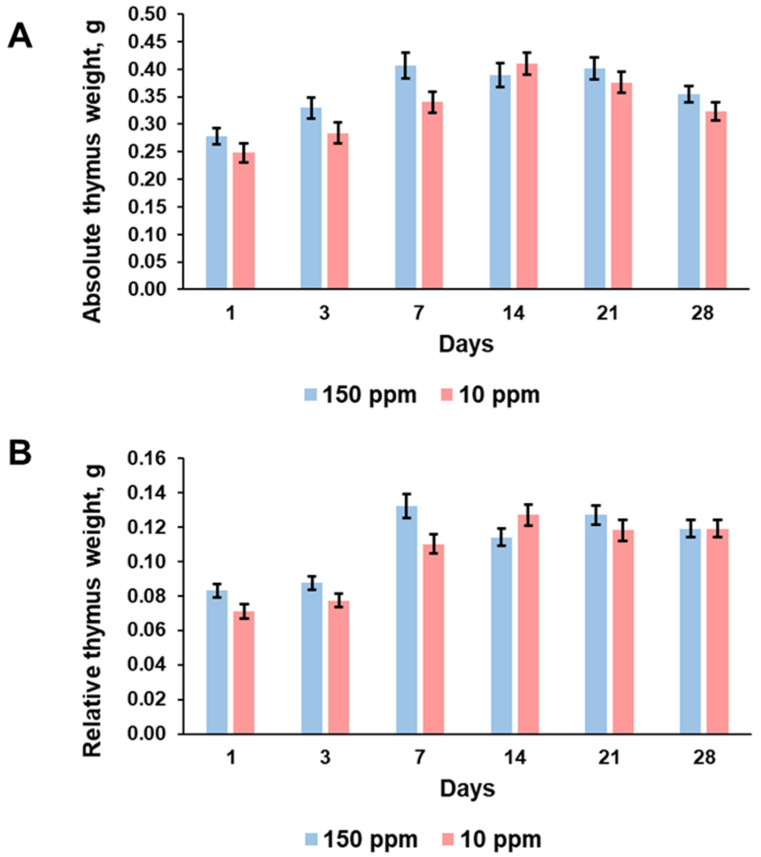
Dynamics of absolute thymus weight (**A**) and relative thymus weight (**B**) of the control rats (150 ppm) and the rats that consumed deuterium-depleted water (10 ppm). Data are shown as mean ± S.E.M.

**Figure 4 biomedicines-12-00956-f004:**
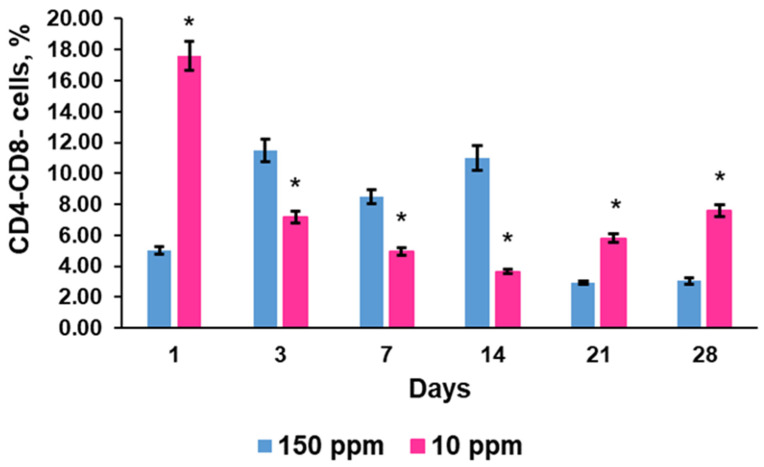
Dynamics of double-negative CD4^−^CD8^−^ progenitor cells in the thymus of the control rats (150 ppm) and the rats that consumed deuterium-depleted water (10 ppm). Data are shown as mean ± S.E.M. *p* < 0.05 compared to the control (*).

**Figure 5 biomedicines-12-00956-f005:**
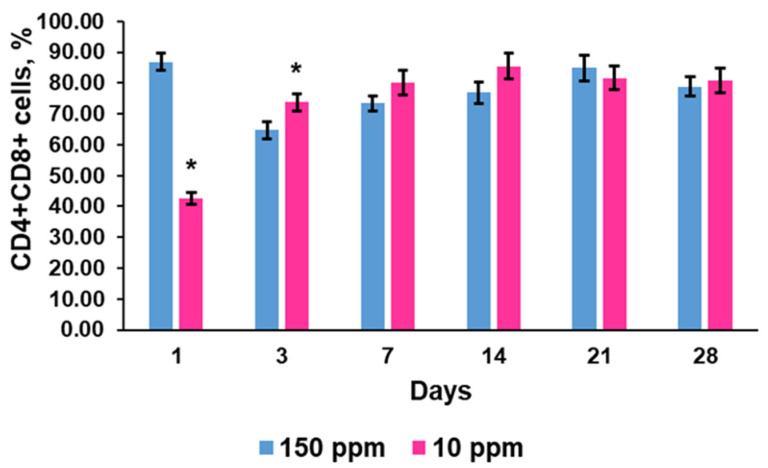
Dynamics of double-positive CD4^+^CD8^+^ cells in the thymus of the control rats (150 ppm) and the rats that consumed deuterium-depleted water (10 ppm). Data are shown as mean ± S.E.M. *p* < 0.05 compared to the control (*).

**Figure 6 biomedicines-12-00956-f006:**
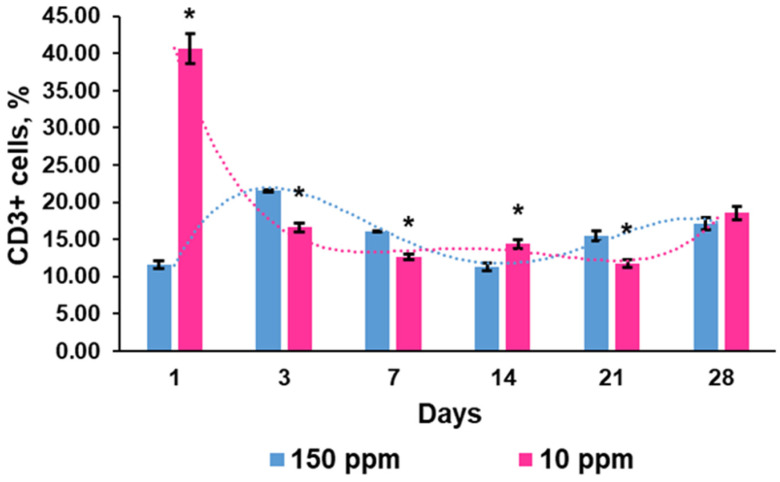
Dynamics of CD3^+^ lymphocytes in the thymus of the control rats (150 ppm) and the rats that consumed deuterium-depleted water (10 ppm). Data are shown as mean ± S.E.M. Dotted lines demonstrate trends in the dynamics of CD3^+^ cell population in the thymus. *p* < 0.05 compared to the control (*).

**Figure 7 biomedicines-12-00956-f007:**
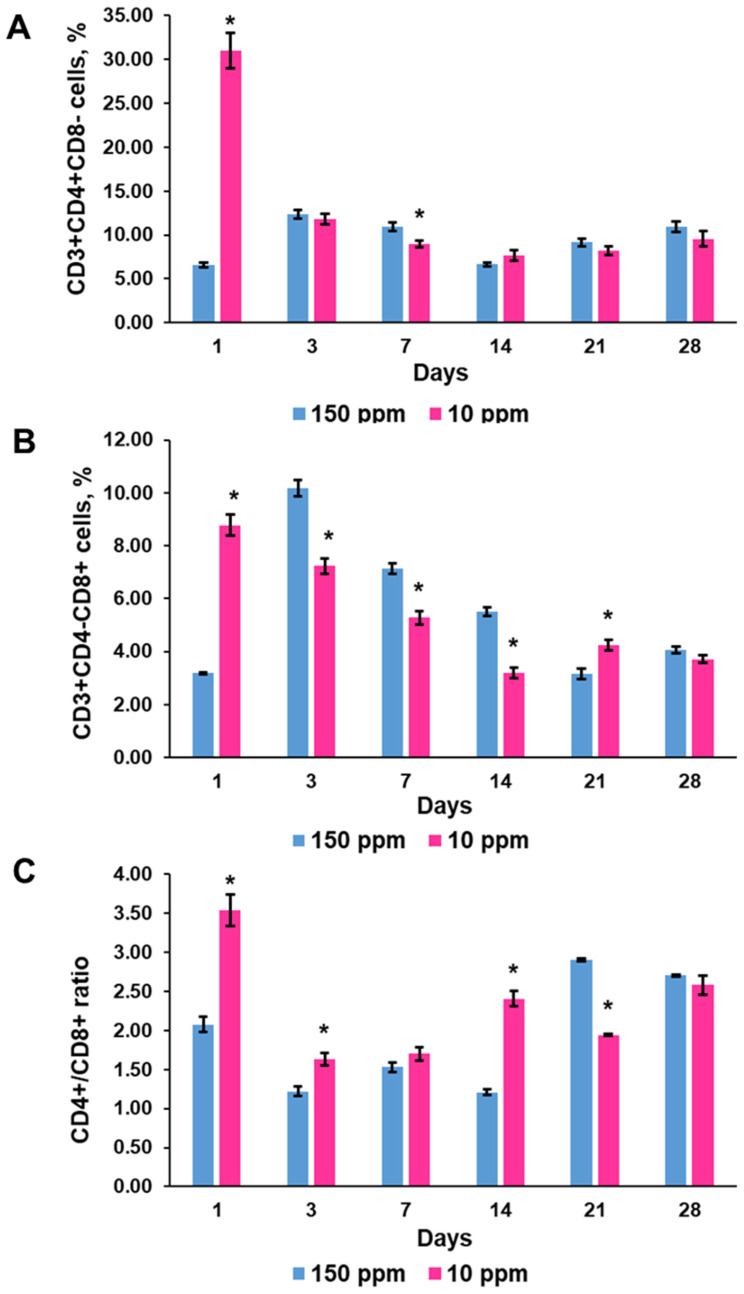
Dynamics of CD3^+^CD4^+^ T-helpers (**A**), CD3^+^CD8^+^ T-cytotoxic lymphocytes (**B**), and T-helper/T-cytotoxic lymphocyte ratio (**C**) in the thymus of the control rats (150 ppm) and the rats that consumed deuterium-depleted water (10 ppm). Data are shown as mean ± S.E.M. *p* < 0.05 compared to the control (*).

**Figure 8 biomedicines-12-00956-f008:**
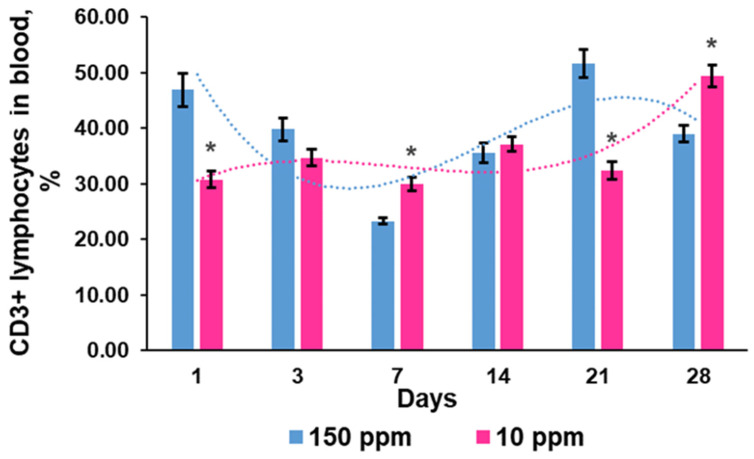
Dynamics of CD3^+^ lymphocytes in the blood of the control rats (150 ppm) and the rats that consumed deuterium-depleted water (10 ppm). Data are shown as mean ± S.E.M. Dotted lines demonstrate trends in the dynamics of the CD3^+^ cell population in the blood. *p* < 0.05 compared to the control (*).

## Data Availability

The data presented in this study are available from the corresponding author upon reasonable request.

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
