# Peer review of "Effects of Deuterium Depletion on Age-Declining Thymopoiesis In Vivo"

_biomedicines, 2024, doi:10.3390/biomedicines12050956_

Round 1
Reviewer 1 Report
Comments and Suggestions for Authors
Rat setting: The concept of following the effect of deuterium depletion on the thymus is novel. However, the current manuscript is highly descriptive, sporadically even speculative and totally lacks functional read-outs. It would be of extreme use to assess functional alterations of thymocytes / fresh naive T-cells that occur due to deuterium depletion.
Human setting: With smaller variation, but natural waters also show variance for deuterium content. Comparative studies between indigenous population and tourists could perhaps partly inforce rat data to enhance the human relevance of data. Even human studies are foreseeable as deuterium depletion does not impose a health hazard.
Comments on the Quality of English Language
English text quality is not homogenous throughout the text. It appears that multiple authors wrote the manuscript with different levels of using English as a foreign language.
Author Response
Dear Reviewer! We thank you for your work in reviewing our manuscript. Corrections and additions are highlighted in red.
Comment:
Rat setting: The concept of following the effect of deuterium depletion on the thymus is novel. However, the current manuscript is highly descriptive, sporadically even speculative and totally lacks functional read-outs. It would be of extreme use to assess functional alterations of thymocytes / fresh naive T-cells that occur due to deuterium depletion.
Human setting: With smaller variation, but natural waters also show variance for deuterium content. Comparative studies between indigenous population and tourists could perhaps partly inforce rat data to enhance the human relevance of data. Even human studies are foreseeable as deuterium depletion does not impose a health hazard.
Response:
We did not set out to study the functional indicators of T cells, since our main aim was to find out whether a decrease in the deuterium content in the body is capable of stopping age-related involution and activating the generation and maturation of T cells.
We clarified the purpose of the study, indicating that it was devoted to the study of morphogenetic processes in the thymus.
We also have updated the Discussion chapter and included limitations of the study.
Reviewer 2 Report
Comments and Suggestions for Authors
Dear Authors,
The manuscript entitled "Effects of deuterium depletion on age-declining thymopoiesis in vivo" describes the potential use of deuterium to enchance the thymopoiesis in rats. However, i have major concerns for the current manuscript, before further processed. Below you can find my comments.
1) The idea of the manuscript is very optimistic and impressive indeed, and will be of great interest for the scientistis in the field. However, the evidence provided by the authors regarding the potential effect of deuterium cannot be supported by the provided results.
2) The authors should also evaluate thymus sensitive markers, including TSH, T3 and T4 for establishing better its function after the deuterium application.
3) Moreover labeled CD3, CD4-, CD8-, CD3+CD4+CD8-, CD3+CD4-CD8+, CD3-CD4+CD8+, should also be used in order to monitor live their function in thymus.
4) In the introduction section, the sentences ". Chemical reactions involving protium...putative regulator on metabolic and physiological processes." are not necessary and it could be better to be removed.
5) In the discussion section, the authors just repeated the findings of their experimental approach. The authors should discuss their results with similar studies.
6) What will be the future perspectives of this study and also how can deuterium can be applied in humans.
7) Also what are the limitations of the study.
Author Response
Response to Reviewer
Dear Reviewer! We thank you for your work in reviewing our manuscript. We have made all necessary corrections to the text of the article. Corrections and additions are highlighted in red.
Comment 1:
The idea of the manuscript is very optimistic and impressive indeed, and will be of great interest for the scientists in the field. However, the evidence provided by the authors regarding the potential effect of deuterium cannot be supported by the provided results.
Response 1:
The present results show an increase in number of thymoblasts and their intensified differentiation after 4 weeks of deuterium depletion. It allows to consider deuterium depletion as a potential approach to restore thymopoiesis in adults.
Comment 2:
The authors should also evaluate thymus sensitive markers, including TSH, T3 and T4 for establishing better its function after the deuterium application.
Response 2:
We have already published the data on thyroid function and mentioned them in the Discussion.
Comment 3:
3) Moreover labeled CD3, CD4-, CD8-, CD3+CD4+CD8-, CD3+CD4-CD8+, CD3-CD4+CD8+, should also be used in order to monitor live their function in thymus.
Response 3:
The aim of our research was primarily to assess morphogenetic parameters in the thymus. We quantified negative, double- and single-positive cells to evaluate how the blasts differentiate into T cells. That is why we used flow cytometry. Additional quantification of T cells in the blood allowed to assess process of migration.
Comment 4:
In the introduction section, the sentences ". Chemical reactions involving protium...putative regulator on metabolic and physiological processes." are not necessary and it could be better to be removed.
Response 4:
We modified the sentence.
Comment 5:
In the discussion section, the authors just repeated the findings of their experimental approach. The authors should discuss their results with similar studies.
Response 5:
Our study focuses on evaluation of morphogenetic processes. Since the results may be interesting not only for specialists in cytology and cell biology, we considered it necessary to provide the readers with a detailed explanation of cell differentiation and migration. In the discussion chapter we also mentioned some papers which allow to assume possible mechanisms underlying the revealed changes. We agree with your comment that we should discuss our results with similar studies, but we can not provide it since there are no similar studies yet.
Comment 6:
What will be the future perspectives of this study and also how can deuterium can be applied in humans.
Response 6:
We suppose that the present results will create a basis for the regulation of thymopoiesis in pathological processes.
Comment 7:
Also what are the limitations of the study.
Response 7:
The limitations of the study include age of the rats (young adults). We have added the limitations to the text.

Round 2
Reviewer 2 Report
Comments and Suggestions for Authors
The authors have succesfully addressed the majority of my comments. However, I do feel that 2 of my comments require further clarification. Please check the comments below.
1. Please provide evidence regarding the quantification of thymus function markers such as TSH, T3 and T4 in the submitted manuscript.
2. Please also perform live monitoring of the T cells subpopulation
3.The discussion needs further refinement, the authors just present their results. The authors should further discuss and compare their results with similar studies.
Round 3
Reviewer 2 Report
Comments and Suggestions for Authors
Dear Authors,
You have addressed my concerns, however i do not feel that the results of this study may support the main concept of the article. I will let the Academic Editor to decide.
Author Response
Dear reviewer! Thank you for your comments. We believe that the study performed fully corresponds to the stated objectives, and the conclusions objectively reflect the results of the study.